# Teeth Enamel Ultrastructural Analysis of Selected Equidae Taxa

Vitalii Demeshkant [1,*] , Michał Biegalski [2] and Leonid Rekovets [3,4]

1 Department of Experimental Biology, Wrocław University of Environmental and Life Sciences, ul. Norwida 27b, 50-375 Wrocław, Poland
2 Faculty of Biological Sciences, University of Wrocław, plac Uniwersytecki 1, 50-137 Wrocław, Poland
3 Division of Vertebrate Ecology and Paleontology, Wrocław University of Environmental and Life Sciences, ul. Chełmonskiego 38c, 51-630 Wrocław, Poland
4 National Science and Natural History Museum of the National Academy of Sciences of Ukraine, Bohdana Khmel'nyts'koho St. 15, 01030 Kyiv, Ukraine
* Correspondence: vitalii.demeshkant@upwr.edu.pl

**Abstract:** This paper presents historical and evolutionary insights into the "tarpan" group of small horses by examining molar tooth enamel ultrastructure. Mathematical methodologies were applied to enhance the analysis. Tooth enamel from species such as *Equus gmelini* (tarpan), *E. latipes*, and *E. hydruntinus* from Pleistocene Ukrainian localities, *E. przewalskii* from the Chornobyl Exclusion Zone in Ukraine, and *E. caballus* form *sylvaticus* (Polish konik) from Roztocze National Park, Poland, underwent scanning microscope examination. Measurements of enamel structures, including prisms (PE) and interprismatic matrix (IPM), were conducted, with the K-index used as their ratio, categorized by enamel type (I, II, III). The findings confirmed that the crystal structures of enamel in these horse groups vary based on genus evolution, diet, and environmental conditions, shaping the enamel's morphological features. Through analysis, clusters were identified, allowing for potential reconstructions of relationships among study groups. The results revealed distinct differences between species, enabling their classification within an established phenogram. Two primary clusters emerged: one consisting of extinct small horse forms from diverse localities and another grouping modern forms. Notably, the Late Pleistocene European species *E. latipes* showed close affinities to the latter cluster.

**Keywords:** horse; teeth; enamel; morphology

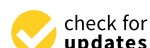



## 1. Introduction

The evolutionary trajectory of horses, particularly within the Eurasian group, remains a subject of ongoing debate. Recent years have witnessed an exploration of evolutionary and phylogenetic matters through the integration of morphological characteristics and DNA analysis of osteological remains. This approach has enabled the precise determination of phylogenetic relationships among horse species. Another avenue of investigation involves examining the broader functional system in relation to its constituent elements. The inclusion of tooth enamel in such research, along with the application of statistical analyses, represents an important step toward unraveling the intricate relationships between species within the Equidae family. This is evidenced by a number of publications in recent years aimed at solving the problems of systematics and phylogeny of Old World horses on the basis of additional osteological features.

Herbivorous terrestrial organisms offer a compelling case study for observing adaptogenesis and coevolutionary processes. Among these organisms, the phylogenesis of horses stands out as an exemplary demonstration of these phenomena [1]. The wear patterns observed in the teeth of ungulate species, which are frequently attributed to the presence of phytoliths, serve as a notable illustration of the interdependent evolution between ungulates and plants [2]. The ridges and folds of tooth enamel, as well as dentine, have a significant impact during forage grinding.

Tooth morphology, particularly tooth enamel, in horses has evolved towards optimizing the efficiency of cellulose grinding and promoting a healthier gut microbiota [3].

The Equidae that were analyzed are represented by several different species, with their teeth having been subject to partial examination in prior studies [4].

The aim of this study was to examine the developmental history and evolution of both extinct and modern small horse species in a conditionally separated group of "tarpan" horses. For this purpose, the ultrastructure of the tooth enamel of such extinct forms as *Equus latipes*, *E. hydruntinus*, and *E. gmelini* from different Pleistocene and Holocene localities, as well as modern *Equus przewalskii* and *Equus caballus* (Polish konik), was studied. The research hypothesis assumes that the study of the ultrastructure of tooth enamel in various species of Equidae will lead to the substantiation of additional morphological features in order to use them in taxonomy and systematics. This investigation encompassed a comparative analysis of data pertaining to the ultrastructure of molar enamel, wherein mathematical techniques were employed for the systematic examination and subsequent interpretation of the results. The data obtained allowed us to identify morphological dependencies between the studied groups and schematically present the results as morphogenesis.

## 2. General Characteristics of Tooth Enamel

The process of enamel formation, occurring concurrently with tooth development during embryogenesis, encompasses distinct growth phases and plays a fundamental role in morphogenesis and evolution of organisms. Enamel represents an adaptive and enduring structure that underlies its diverse manifestations and variations, contingent upon the arrangement of its crystal layers. The basic structural components comprise hydroxyapatite crystals and the interprismatic matrix (IPM) structure, which exhibit specific spatial arrangements in relation to one another [5,6].

Enamel structures display divergences within the analyzed forms, and our study demonstrates their taxonomic importance, especially for the Equidae family. In the case of horses, the tooth enamel predominantly displays a radial structure, which can be classified into three distinct types (I, II, III) [7]. The first enamel type is consistently positioned closer to the enamel–dentine junction (EDJ). It constitutes approximately 40% of the overall tooth enamel layer and is characterized by rows (strands) of prisms (PE) and matrix structures (IPM) arranged in parallel and sequential fashions (Figure 1). The prisms tend to aggregate in clusters, resembling lens-like structures (isolated prisms being a rare occurrence), and are consistently oriented in a tilt towards the EDJ. The rows of IPM in type I molar teeth display a weaker or less pronounced crystal structure, with a tendency to divide the rows towards the outer enamel surface (OES). These IPM rows are wider and thicker than the PE rows. Additionally, the thickness of both PE and IPM rows proportionally varies as one approaches the enamel–dentine junction (EDJ).

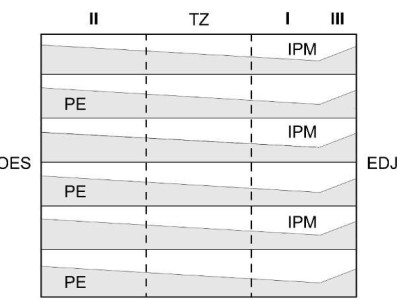 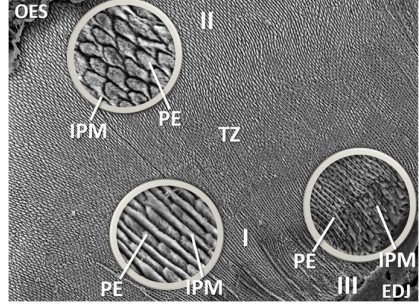

**Figure 1.** Diagram of the enamel structure of molars. I, II, III—types of enamel structures. TZ—transitional zone. OES—outer enamel surface. EDJ—enamel-dentine junction. PE—enamel prisms. IPM—interprismatic matrix.

Type II enamel is located in closer proximity to the border of the outer enamel surface (OES). It constitutes approximately 40–45% of the total tooth enamel surface area. Similar to type I enamel, it is composed of prisms (PE) and interprismatic matrix (IPM) structures, arranged in relatively parallel rows. Type II enamel is characterized by a distinct horseshoe-shaped arrangement of the prismatic enamel (PE) group, indicating a pronounced tilt towards the outer enamel surface (OES) borders. In type II enamel, the PE prisms are notably thicker than the interprismatic matrix (IPM), with the latter being scarcely discernible.

Type III enamel is closely associated with dentine ameloblasts and the enamel formation process. It constitutes approximately 5–8% of the total tooth enamel surface area. The prisms (PE) in type III enamel form clusters (strands) that are either round or elongated, exhibiting a wider configuration. Type III enamel exhibits a distinct feature with the absence of clearly defined IPM rows, setting it apart from the other types. Its morphology and functional significance are relatively less emphasized compared to the preceding types. The boundaries between the different enamel types exhibit a transitional nature, particularly observed in the transitional zone (TZ), which is recognized for its indistinct characteristics of boundaries [4].

## 3. Materials and Methods

The molar teeth of fossil forms, including *Equus latipes*, *Equus gmelini*, and *Equus hydruntinus*, from various Holocene and Late Pleistocene locations, as well as contemporary specimens of *Equus caballus* form *sylvaticus* (Polish konik) from Roztocze National Park in Poland and *Equus przewalskii* from the Chornobyl Exclusion Zone in Ukraine, were used in this study. The illustration depicting the map of the research site locations from which the study material was obtained can be found in a previous article (Figure 1 in [4]). As a component of the research methodology, a total of 28 molar teeth, including 4 from each form, underwent analysis using scanning electron microscopy (SEM).

The species *Equus latipes* is recognized as a specific form found in the periglacial zone of Europe, particularly from the Mizyn locality of the Late Pleistocene period (19.6 thousand years ago) in northern Ukraine, near the Desna River. The species *Equus gmelini*, commonly known as the tarpan, had a wide distribution across the steppes of Europe and Asia. The tarpan existed from the early Holocene period until the late nineteenth century. Fossil remains of the tarpan have been discovered at Late Pleistocene localities in Myrne, Hirzhevo, and Kamiana Mohyla in Ukraine. The Myrne locality is recognized as an archaeological locality of the Pleistocene–Holocene epoch. Its late Mesolithic phase is estimated to be approximately 8.5 to 9 thousand years old. The Hirzhevo locality is representative of the early Neolithic period, specifically the first half of the Holocene, with an estimated age of approximately 7 thousand years. The mammalian remains from the Kamiana Mohyla locality include osteological remnants of tarpans belonging to various periods of the Holocene [8]. Discussions regarding the taxonomic designation of the extinct wild horse of Europe remain pertinent [9]. In the present work, we adhere to the name *Equus gmelini*, which was used in the previous study [4].

The extinct European donkey *Equus hydruntinus* is known from the Kabazi 2 locality in Crimea, Ukraine. It occupied the southern and, in part, central regions of Europe, in addition to the Middle East and Iran, during the latter half of the Pleistocene [10].

The Polish konik, *Equus caballus* L. 1758, is known as a hybrid resulting from the crossbreeding of the tarpan and domesticated *Equus caballus*. This hybridization occurred during the nineteenth century when domesticated horses returned to the wild and interbred with the remaining tarpan populations. The Polish konik exhibits notable differences from the steppe tarpan found in the southern part of Eastern Europe. According to Jaworska et al. [11], the Polish konik form bears the closest resemblance to populations of this species from the Białowieża Forest region in Poland. These distinctions highlight the unique characteristics and genetic makeup of the Polish konik population in comparison to the steppe tarpan found in other regions of Eastern Europe.

*Equus przewalskii* Poliakov 1881 is a species with a well-documented history, as evidenced by numerous publications. However, its taxonomy and systematic classification remain subjects of ongoing debate. Three primary perspectives exist regarding the nomenclature of this taxon: considering it as an independent species, *Equus przewalskii*, as a subspecies of the tarpan (*Equus gmelini przewalskii*), or as a subspecies of the horse (*Equus caballus przewalskii*).

The preparation of samples for the study was carried out in accordance with generally accepted methods and described in detail in a previous work [4]. The results of the experiments, in the form of computer-generated images of tooth enamel ultrastructure obtained through a Zeiss® (Jena, Germany) EVO LS 15 scanning electron microscope (SEM), were subjected to morphometric analysis of the key structural elements, namely prisms (PE) and interprismatic matrix (IPM). Three types of enamel (I, II, III) were identified [7] and their prism (PE) and interprismatic matrix (IPM) band widths were measured (Figure 1). The widths varied along the enamel–dentine junction (EDJ) and outer enamel surface (OES). The K-index (ratio of PE to IPM width) was used to determine the dynamics of width variations. Additionally, the Prisms Inversion Index (PII) was introduced. It represents the ratio of the K-index for type I to the K-index for type II, expressed as a percentage.

In summary, the objective of the study was to conduct a comparative analysis of the data on enamel ultrastructure in selected groups of horses using experimental data and previously published materials [4]. Particular attention was paid to the morphological features of the enamel of the species *Equus latipes*, which have not been previously described. The use of a test approach to the comparative processing of data on different types of enamel allowed us to identify the leading morphological and functional feature, namely type I enamel, which was used in the morphogenesis scheme.

We used a combination of statistical methods to assess whether there were any differences between the different groups of horses studied. To begin analysis, a normality test was needed to see whether K-index had a normal distribution. This means that both the asymmetry index and kurtosis of the samples are equal to 0.

$$\text{Asymmetry}$$
$$\gamma\_1 = (E(X - EX)^3)/(\text{Var}(X)^{(3/2)})$$
$$\text{Kurtosis}$$
$$\gamma\_2 = (E(X - EX)^4)/(\text{Var}(X)^2) - 3$$

$$\text{Test statistic}$$
$$W = (\sum\_(I = 1)^{(n/2)} \llbracket a\_i(n)(X\_{(n-I+1:n)} - X\_{(i:n)}) \rrbracket)^2 / (\sum\_(I = 1)^n (X\_i - X_\circledcirc)^2)$$

The null hypothesis $H_0$ is discarded if $W < w\alpha(n)$, while $w\alpha(n)$ is the corresponding critical value of the S-W distribution. R Statistics 4.0.3 was used for data analysis and visualizations as part of the research methodology.

Two non-parametric statistical tests were utilized in our analysis. The first, the Kruskal–Wallis rank sum test, involves ranking all data from smallest to largest, summing ranks within subgroups, and calculating the H statistic:

$$H = 12/N(N+1) \sum (R\_i^2)/n\_i - 3(N+1)$$

The second test, Nemenyi's all-pairs rank sum test, is suitable for all-pairs comparisons in one-factorial layouts with non-normally distributed residuals. A total of $m = k(k-1)/2$ hypotheses can be tested. The null hypothesis $H\_{ij}:\theta\_i(x) = \theta\_j(x)$ is tested in the two-tailed test against the alternative $A\_{ij}:\theta\_i(x) \neq \theta\_j(x)$, $i \neq j$. Let $R\_{ij}$ be the rank of $X\_{ij}$, where $X\_{ij}$ is jointly ranked from 1, 2, …, N, $N = \sum\_(I = 1)^k n\_i$, then the test statistic under the absence of ties is calculated as $t\_{ij} = ((R\_j)_\circledcirc - (R\_i)_\circledcirc)/(\sigma\_R(1/n\_i + 1/n\_j)^{(1/2)})$ $(i \neq j)$ with $(R\_j)_\circledcirc, (R\_i)_\circledcirc$ the mean rank of the i-th and j-th group and the expected variance as $\sigma\_R^2 = N(N+1)/12$. A pairwise difference is significant if $|t\_{ij}/\sqrt{2} > q\_{kv}$ with k the number of groups and $v = \infty$ the degree of freedom. A modified approach for Nemenyi's test was used in the presence of ties for $N > 6$, $k > 4$ provided

that the K-W test indicates significance [12]. In the presence of ties, the test statistic is corrected according to $(t\_ij)^{\wedge} = t\_ij \backslash /C$, with $C = 1 - (\sum\_(I = 1)^{\wedge}r\_t\_i^{\wedge}3 - t\_i)/(N^{\wedge}3 - N)$.

The distribution is provided from the studentized range distribution, and the *p*-value is computed on the basis of the following formula: $Pr(t\_(ij\sqrt{2}) \geq q\_k\infty\alpha \backslash |mathrmH) = \alpha$.

## 4. Results

In this study, the teeth of *Equus latipes* were discovered in the Late Pleistocene deposits in Mizyn, Ukraine, representing one of the species within the mammoth fauna of the periglacial zone of Europe, which was well adapted to the conditions of the steppe tundra [13]. Tooth enamel had a different and more archaic structure in comparison to other forms. This enamel has relatively wide enamel prisms (PE) in type I, and its crystalline structures are not arranged in bundles and are almost parallel to the OES boundary surface near the chewing surface. Type III enamel of *Equus latipes* rarely occurs near the EDJ. The most important feature is inverted structures of IPM and PE of types I and II, which in total make up 95% of the whole tooth enamel width (Figure 2). The images illustrating the ultrastructure of the tooth enamel of the forms studied are published in an earlier article (Figures 4–9 in [4]).

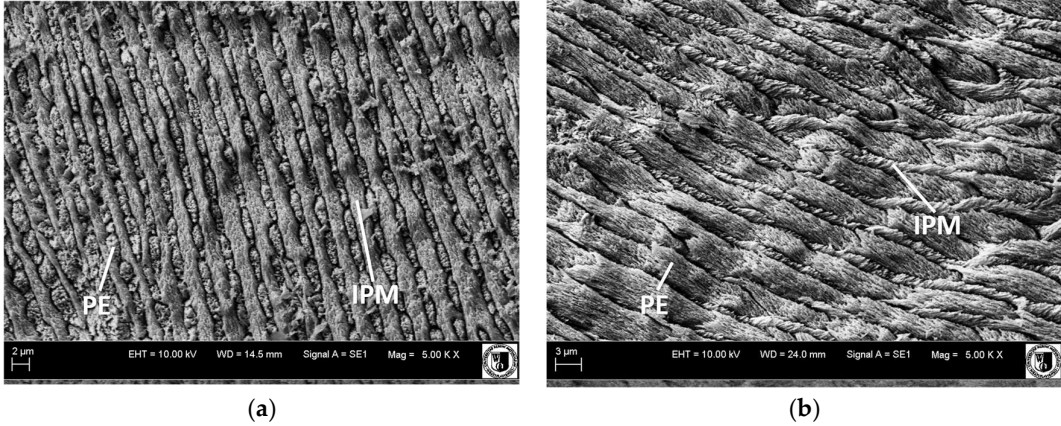

(**a**)　　　　　　　　　　　　　　(**b**)

**Figure 2.** (**a**) Type I tooth enamel structure of *Equus latipes* from Mizyn locality; (**b**) type II tooth enamel structure of *Equus latipes* from Mizyn locality.

The species *Equus gmelini* from the examined localities retains a relatively primitive structure of molar tooth enamel. This means that it has a poorly developed type III enamel, wide PE and IPM bands, a clearly visible TZ, and a horseshoe-like form of PE prisms in type II enamel. Enamel of the species from the Myrne locality (unlike others) has prism decussation in the TZ, in the type I enamel from the Hirzhevo locality the PE and IPM rows are quite parallel, and in that from the Kamiana Mohyla locality the PE and IPM rows have a weak prism structure in the TZ (Figures 4–6 in [4]).

In *Equus hydruntinus*, both upper and lower molars had well-developed wide tooth enamel strands along the perimeter, which proves its separateness as a species in comparison to others. Between borders, OES and EDJ tooth enamel can be divided into three equal parts: type I near the EDJ, type II near the OES, and a wide TZ in between them. Each of the parts has its own characteristics and makes up around 30% of tooth enamel width. What distinguishes molar teeth of *Equus hydruntinus* from other tarpan horses is that there is almost no type III tooth enamel and little development of a type II wide TZ (Figure 9 in [4]). Such features distinguish this species from others and confirm its phylogenetic isolation [14]. This species is morphologically similar to small tarpan horses which were adapted to arid steppe and available food. Due to such conditions despite different ancestry, they share morphological similarities in tooth and tooth enamel features with the above species.

*Equus* sp. Myrne was designated as *Equus latipes* in the collection materials. Studies conducted confirmed that these remains do not belong to the species but retain morphological differences that made it possible to designate them using open nomenclature (*Equus* sp.) and to mark their similarity to *Equus* of the *caballus* group [15]. The *Equus* sp. form, unlike *E. latipes*, has more developed type III enamel, and in I type the PE bands (prisms) are more arranged in parcels and they are thicker than the IPM prisms. In the form of *Equus* sp., in the middle part of the enamel band between the boundaries of the EDJ and OES, a narrow band of wave-like structures (transition zone, TZ) is often observed, which are poorly filled with prisms and more appropriate for *Equus gmelini*. As a whole, the structure of the enamel of *Equus* sp. from the Myrne locality retains some morphological similarity to tarpans from other localities.

Polish konik, *Equus caballus*, similar to other forms, has two dominant types types I and II. Type I is characterized by noticeable IPM of a linear structure, in between which PE prisms are located. Type II has both tightly layered elliptic and horseshoe-shaped prisms. IPM rows are less developed. The exception is that PE and IPM are angled in opposing ways: the first towards the EDJ and the second towards the OES. According to Rensberger and Koenigswald [5], it increases mechanical hardness of the enamel. Decussations are clearly visible in the TZ (Figure 10 in [4]).

In *Equus przewalskii*, type I makes up 70% of the width of the tooth enamel. It is built from parallel rows of IPM and prisms of PE, which have identical thickness. IPM is grainy and noticeably 30° tilted towards the OES border, where it has a horseshoe shape, like in type II. Unusual are constant wedge-like structures of types I and II in the TZ, not found in any other Equidae. Analysis of data on the bone morphology of Przewalski's horse indicated similarity to domesticated forms from east Kazakhstan that had gone wild. Molecular and genetic data imply its close relatedness to *Equus gmelini* and *Equus caballus* [9]. The comparative analysis was also carried out according to the structure of the enamel (Figure 11 in [4]). The enamel of this species differs from that of other horses of the tarpan group by the wedge-like entry of types I and II in the transition zone.

The morphometric analysis yielded tables and corresponding graphical representations that, following suitable mathematical elaboration, facilitate the comparative characterization of acquired data across diverse taxa spanning various geological epochs. These findings enable the derivation of comprehensive summary data in the form of graphical clusters. The data for type I enamel, including mean K-index values and their corresponding standard deviations, are summarized below. For type I enamel, K-index values are almost always less than 1, which is characteristic of most of the forms studied (Table 1).

**Table 1.** Descriptive statistics of type I enamel.

| Species and Location | Mean K-Index Value | Standard Deviation of K-Ratio |
|---|---|---|
| *E. gmelini* Hirzhevo | 1.08 | 0.20 |
| *E. przewalskii* Chornobyl | \|0.40\| | 0.16 |
| *E. caballus* Roztocze NP | 0.81 | 0.12 |
| *E. gmelini* Kamiana Mohyla | 0.76 | 0.16 |
| *Equus* sp. Myrne | 0.76 | 0.60 |
| *E. gmelini* Myrne | 0.52 | 0.23 |
| *E. latipes* Mizyn | 0.49 | 0.10 |
| *E. hydruntinus* Kabazi 2 | 1.04 | 0.26 |

If the K-index values are within 1 (with a slight deviation), it means that PE and IPM have equal width. The higher the K-index values, the greater the difference between the width of PE and IPM prisms.

Starting from the data obtained, it is obvious that, within the limits of type I, the bands of PE and IPM prisms have almost the same width in *E. hydruntinus* and *E. gmelini* from Hirzhevo. The bands differ most by width in *E. latipes* and *E. gmelini* from Myrne (Figure 3).

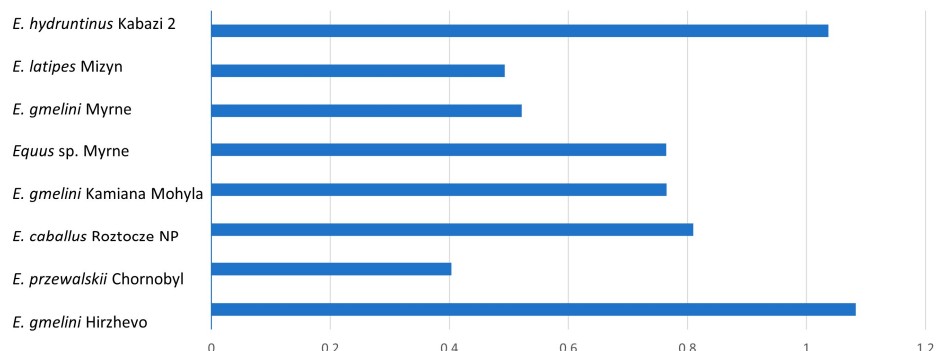

**Figure 3.** Mean dimension of K-index for type I tooth enamel.

For type II enamel, the K-index will always be greater than 1. The higher the K-index values, the greater the difference between the width of PE and IPM prisms (Table 2).

**Table 2.** Descriptive statistics of type II enamel.

| Species and Location | Mean K-Index Value | Standard Deviation of K-Ratio |
|---|---|---|
| *E. gmelini* Hirzhevo | 2.38 | 0.66 |
| *E. przewalskii* Chornobyl | 1.72 | 0.52 |
| *E. caballus* Roztocze NP | 2.37 | 0.63 |
| *E. gmelini* Kamiana Mohyla | 3.75 | 1.70 |
| *Equus* sp. Myrne | 3.64 | 0.60 |
| *E. gmelini* Myrne | 1.56 | 0.40 |
| *E. latipes* Mizyn | 3.16 | 0.87 |
| *E. hydruntinus* Kabazi 2 | 3.88 | 0.81 |

The largest differences in these indices are in *E. hydruntinus* (3.88), *E. gmelini* Kamiana Mohyla, *Equus* sp. Myrne, and *E. latipes* (Table 2, Figure 4). This indicates a rather weak development of IPM bands.

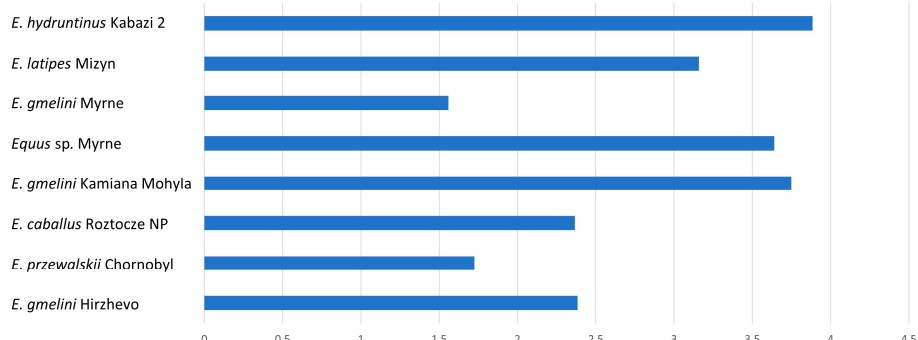

**Figure 4.** Mean dimension of K-index for type II tooth enamel.

Similar characteristics to type II are found in type III enamel, which differs in having quite wide PE prisms and poorly developed IPM (Table 3). We observe a change in the width of PE and IPM in different forms from different geological times. In older forms such as in *E. hydruntinus*, we see small differences in width between PE and IPM, while in younger forms the trend is the opposite (Figure 5). Also confirmed earlier was the dynamics of morphological changes, concerning the dynamics of changes in the direction of inclination of prisms towards the OES boundary. This is related to the type of food and to the change in environmental conditions [4].

**Table 3.** Descriptive statistics of type III enamel.

| Species and Location | Mean K-Index Value | Standard Deviation of K-Ratio |
|---|---|---|
| *E. gmelini* Hirzhevo | 2.32 | 0.46 |
| *E. przewalskii* Chornobyl | 1.81 | 0.45 |
| *E. caballus* Roztocze NP | 2.77 | 0.71 |
| *E. gmelini* Kamiana Mohyla | 2.11 | 0.47 |
| *Equus* sp. Myrne | 1.91 | 0.41 |
| *E. gmelini* Myrne | 2.50 | 0.46 |
| *E. latipes* Mizyn | 2.54 | 1.02 |
| *E. hydruntinus* Kabazi 2 | 1.52 | 0.54 |

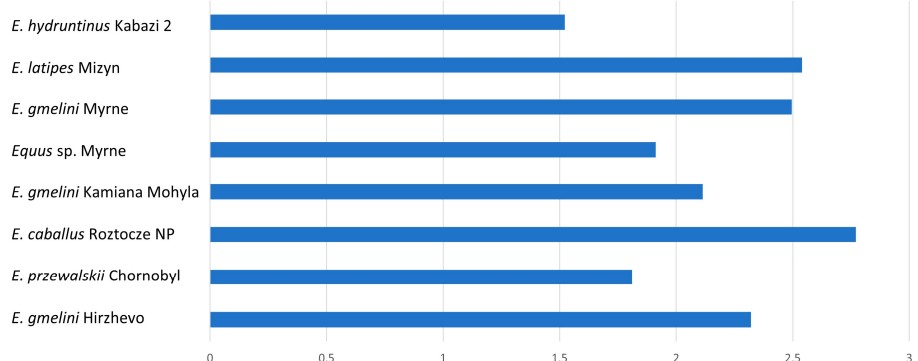

**Figure 5.** Mean dimension of K-index for type III tooth enamel.

For the forms studied, relatively well-developed IPM in *E. przewalskii, Equus* sp., and *E. hydruntinus* (Figure 5) is found.

Morphometric data on the structure of different enamel types were also analyzed to determine trends of change and the interrelationship between the enamel prisms (PE) and interprismatic matrix (IPM) in each enamel type for all the studied forms together. This analysis was aimed at better understanding the evolution of enamel structure and the relationship between different equine species (Figure 6).

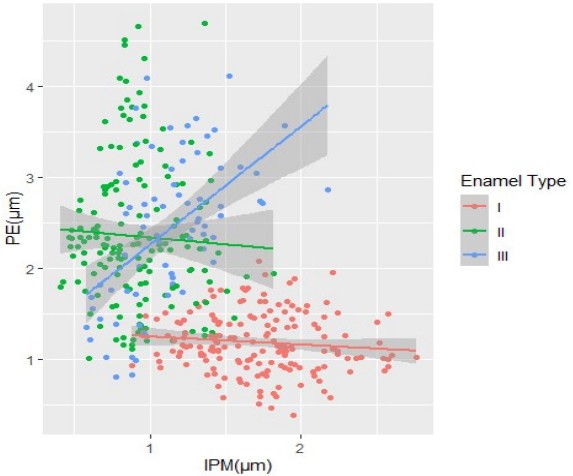

**Figure 6.** Morphometrical measures PE and IPM and tendencies of the change of enamel types.

Graphic data show that IPM and PE bands in the first enamel type maintain a relatively stable structure in width, while in the second enamel type PE prisms show a sharp increase in width. Similar trends can also be observed in the third enamel type (Figure 6).

Morphometric data of enamel types were also analyzed using the Shapiro–Wilk and Kruskal–Wallis tests with the separation of comparative groups (A, B, C). Shapiro–Wilk normality tests were conducted on all three data groups and all proved not to be normally distributed (Table 4).

**Table 4.** Results of the Shapiro–Wilk distribution normality test for enamel types.

| Enamel Type | W | p-Value |
|---|---|---|
| Type I | 0.98 | $3.95 \times 10^{-3}$ |
| Type II | 0.95 | $2.57 \times 10^{-5}$ |
| Type III | 0.96 | 0.01 |

The Kruskal–Wallis test yielded significant results for each group (Table 5). Ranks calculated in this test allowed segregation of species into three groups (A, B, C) (Table 6).

**Table 5.** Kruskal–Wallis test analysis data for different types of enamel.

| Enamel Type | Chi-Squared | p-Value |
|---|---|---|
| Type I | 117.61 | $2.2 \times 10^{-16}$ |
| Type II | 83.58 | $2.56 \times 10^{-15}$ |
| Type III | 26.18 | $4.68 \times 10^{-4}$ |

**Table 6.** Kruskal–Wallis rank values and division into groups in type I enamel.

| Species | Locality | Rank | Group |
|---|---|---|---|
| *E. gmelini* | Hirzhevo | 150.30 | A |
| *E. przewalskii* | Chornobyl | 140.27 | A |
| *E. caballus* | Roztocze NP | 108.85 | B |
| *E. gmelini* | Kamiana Mohyla | 97.34 | B |
| *Equus* sp. | Myrne | 95.17 | B |
| *E. gmelini* | Myrne | 46.47 | C |
| *E. latipes* | Mizyn | 38.95 | C |
| *E. hydruntinus* | Kabazi 2 | 24.15 | C |

Also, the same analysis of group ranking was carried out for type II enamel (Table 7).

**Table 7.** Kruskal–Wallis rank values and division into groups in type II enamel.

| Species | Locality | Rank | Group |
|---|---|---|---|
| *E. hydruntinus* | Kabazi 2 | 121.70 | A |
| *Equus* sp. | Myrne | 115.80 | A |
| *E. gmelini* | Kamiana Mohyla | 110.67 | A |
| *E. latipes* | Mizyn | 97.70 | AB |
| *E. gmelini* | Hirzhevo | 67.52 | BC |
| *E. caballus* | Roztocze NP | 67.00 | BC |
| *E. przewalskii* | Chornobyl | 37.47 | CD |
| *E. gmelini* | Myrne | 28.65 | D |

The morphometric measurements of type I enamel were also analyzed based on the results of the Nemenyi all-pairs rank comparison test (Figure 7). The chart shows that the groups have quite distinct boundaries, which are of a similar focused nature of morphological changes.

To further investigate the Kruskal–Wallis test results, Nemenyi's test was conducted to assess pairwise comparisons. The obtained findings revealed significant differences among various species. Notably, *E. gmelini* specimens from Kamiana Mohyla exhibited distinct variations when compared to *E. hydruntinus* and other *E. gmelini*. Furthermore, notable interactions were observed between *E. przewalskii* and *E. hydruntinus*, as well as between *E. gmelini* from Myrne and *E. latipes* from Mizyn. Remarkably, *E. caballus* specimens from Roztocze NP displayed significant distinctions when compared to *E. latipes* from Mizyn and *E. hydruntinus*. Interestingly, the only two species that did not exhibit significant differences from *E. latipes* from Mizyn were *E. hydruntinus* and *E. gmelini* from Myrne. These outcomes align with the previously established rank-based groups, as outlined below (Table 8).

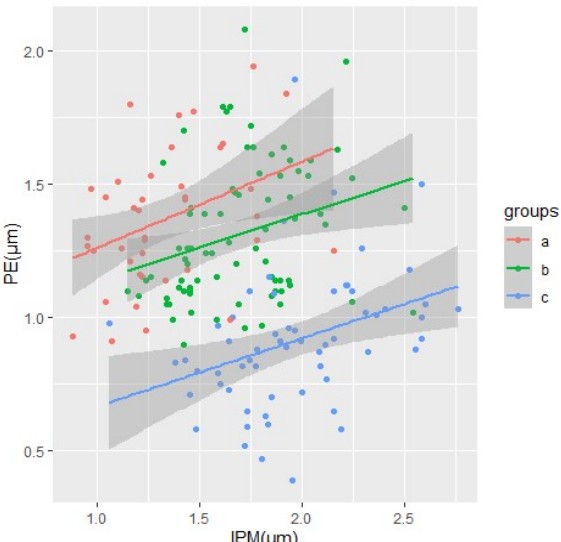

**Figure 7.** Morphometrical measures PE and IPM and tendencies of the change of the type I enamel in ABC groups.

**Table 8.** Nemenyi's All-Pairs Rank Comparison Test for type I enamel.

| Species | *E. przewalskii* Chornobyl | *E. latipes* Mizyn | *Equus* sp. Myrne | *E. gmelini* Hirzhevo | *E. hydruntinus* Kabazi 2 | *E. gmelini* Kamiana Mohyla | *E. gmelini* Myrne |
|---|---|---|---|---|---|---|---|
| *E. latipes* **Mizyn** | $1.8 \times 10^{-8}$ | - | - | - | - | - | - |
| *Equus* **sp. Myrne** | 0.10 | 0.01 | - | - | - | - | - |
| *E. gmelini* **Hirzhevo** | 0.99 | $3.0 \times 10^{-10}$ | 0.02 | - | - | - | - |
| *E. hydruntinus* **Kabazi 2** | $3.83 \times 10^{-11}$ | 0.98 | $3.9 \times 10^{-4}$ | $4.3 \times 10^{-13}$ | - | - | - |
| *E. gmelini* **Kamiana Mohyla** | 0.16 | 0.01 | 1.00 | 0.03 | $2.8 \times 10^{-4}$ | - | - |
| *E. gmelini* **Myrne** | $2.9 \times 10^{-7}$ | 0.99 | 0.06 | $6.6 \times 10^{-9}$ | 0.87 | 0.04 | - |
| *E. caballus* **Roztocze NP** | 0.34 | $2.3 \times 10^{-5}$ | 0.98 | 0.07 | $6.7 \times 10^{-8}$ | 0.99 | $3.0 \times 10^{-4}$ |

In conclusion, it can be noted that all forms except Hirzhevo differ significantly from *Equus latipes* from Mizyn.

A similar analysis was carried out for type II enamel (Table 9). Significant differences were found between the following forms: *E. przewalskii* and *E. latipes*, *E. hydruntinus*, *E. gmelini* from Kamiana Mohyla. Another form with many results is *E. hydruntinus*, being different from *E. gmelini* from Hirzhevo and Myrne, *E. caballus*, and, as mentioned before, *E. przewalskii*. The last form with numerous differences is *E. gmelini* from Myrne, with only three groups not showing significant differences: *E. przewalskii*, *E. caballus*, and *E. hydruntinus*.

A phenogram representing the structural characteristics of type I enamel in small horses, provisionally referred to as the "tarpan" group, was made. It includes five analyzed forms (taxa) from eight historical and modern localities. The common coring of these forms was separated into two separate branches: the forms of the *hydruntinus* group and the *gmelini* group, and the galls—the forms of the *latipes* group and the modern ones. However, in the *Equus gmelini* group, we do not observe monolithicity due to the form from Kamiana Mohyla. This may be related to the dating of the remains—their young geological age. The separation of independent branches of *E. hydruntinus* and *E. latipes* is likely, which may indicate their separate taxonomic status. The similarity between *E. przewalskii* and *E. caballus* is the most justified and confirmed by other studies. It is worth noting that the phenogram is based on data on the first type of enamel, because it was confirmed that

this structure is morpho-adaptively the most convincing in these studies and conclusions. The given phenogram is not a reflection of related relationships, but can be one of the indicators in taxonomic considerations and phylogenetic reconstructions (Figure 8).

**Table 9.** Nemenyi's All-Pairs Rank Comparison Test for type II enamel.

| Species | *E. przewalskii* Chornobyl | *E. latipes* Mizyn | *Equus* sp. Myrne | *E. gmelini* Hirzhevo | *E. hydruntinus* Kabazi 2 | *E. gmelini* Kamiana Mohyla | *E. gmelini* Myrne |
|---|---|---|---|---|---|---|---|
| *E. latipes* **Mizyn** | $1.1 \times 10^{-3}$ | - | - | - | - | - | - |
| *Equus* sp. **Myrne** | $3.0 \times 10^{-6}$ | 0.9239 | - | - | - | - | - |
| *E. gmelini* **Hirzhevo** | 0.45 | 0.45 | 0.03 | - | - | - | - |
| *E. hydruntinus* **Kabazi 2** | $3.1 \times 10^{-7}$ | 0.73 | 0.99 | 0.01 | - | - | - |
| *E. gmelini* **Kamiana Mohyla** | $1.4 \times 10^{-5}$ | 0.99 | 1.00 | 0.06 | 0.99 | - | - |
| *E. gmelini* **Myrne** | 0.99 | $7.7 \times 10^{-5}$ | $9.5 \times 10^{-8}$ | 0.14 | $7.7 \times 10^{-9}$ | $5.0 \times 10^{-7}$ | - |
| *E. caballus* **Roztocze NP** | 0.48 | 0.43 | 0.02 | 1.00 | 0.01 | 0.05 | 0.15 |

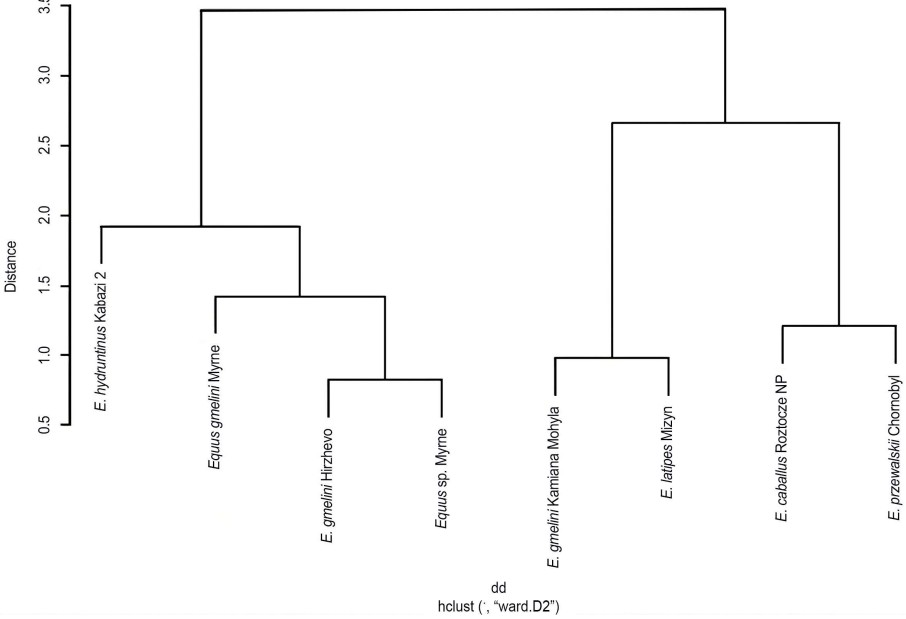

**Figure 8.** Phenogram of similarity among small horses of the tarpan group based on type I enamel structure.

## 5. Discussion

In the evolution of tooth enamel in contemporary Equidae, multiple factors have played pivotal roles, contributing to the present morphological characteristics. The studies and results are in line with the existing data on the structure of enamel in horses by previous authors and deepen our knowledge of its features in small equids of the tarpan group. This study primarily focused on the enamel structure as a morphological feature. The objective was not to address the still-relevant issues of taxonomy and systematics, especially regarding extinct forms such as *E. gmelini*, *E. hydruntinus*, and *E. latipes* [9,15]. It was possible to trace changes in the enamel structure of different taxa of this group and to note its changes and directions in evolution since the late Pleistocene. In our study, the main morphological characteristics of enamel were identified and described in detail, in particular, the ratio of the structure of enamel prisms (PE) and interprismatic matrix (IPM), which were characterized using the K and PII indices. Also, changes in the evolution of the enamel prism structure (PE) were noted, aimed at increasing their width in type I enamel

and a correlated decrease in the width of the IPM. Thus, the peculiarities of the enamel structure and the dependence between PE and IPM in the studied equine forms correspond to the purpose of our research and are supported by mathematical calculations.

This study posited that crystal structures in various horse groups vary based on taxon, genus evolution, diet, and environmental influences, thereby influencing their morphological features. The analysis of selected Equidae specimens revealed both similarities and differences in these crystal structures. According to statistical and group comparative analyses, the distinct groups are *Equus latipes* and *Equus hydruntinus*.

The general assumption of significant differences between species holds true for all three enamel types, with type I having the highest number of pairs that exhibit significant differences. Notably, judging only by means of the K-index, there are relationships between many forms, beginning with *E. gmelini* from Hirzhevo and *E. hydruntinus* from Kabazi 2, which were proven to be significantly different in Nemenyi's test and were ranked as opposites in the K-W test.

Another noteworthy pair consisted of *E. latipes* from Mizyn and *E. przewalskii* from the Chornobyl Exclusion Zone, both of which were positioned on divergent branches of the decision tree. Similarly, despite having a similar mean K-index, this pair was categorized into different groups based on the K-W test, they were significantly different according to Nemenyi's test, and placed opposite each other on the decision tree. The next important difference is to be found between *E. caballus* from Roztocze NP and *E. latipes* and *E. hydruntinus*.

This claim is again supported by both tests and is visualized in the phenogram, where *E. caballus* is on a branch separated from the others. *E. latipes* from Mizyn has the most significant differences between pairs as stated in Nemenyi's test, with only two forms (*E. hydruntinus* and *E. gmelini* Myrne) not being significantly different. This is also reflected in the decision tree. All of those interactions were based on the type I K-index, as this is where most differences were found. Other types of enamel showed some differences between forms. What is worth mentioning is that the K-index of type II enamel of *E. przewalskii* showed differences between both *E. latipes* and *E. hydruntinus* as well as *E. gmelini* from Kamiana Mohyla based on Nemenyi's test. This feature of type I and type II enamel was also examined with conducted tests for *E. caballus* from Roztocze NP. Nemenyi's test showed the difference between *E. hydruntinus* and *Equus* sp. from Myrne. According only to K-index means, it would not be possible to draw such relations and differences between species.

Summary analyses of the data were performed using both the Nemenyi and K-W test, as well as characterizing enamel types I–III. These analyses indicate both differences and similarities between the taxa represented in the morphogenetic diagram. It can be concluded that the small horses of the tarpan group are divided into two main groups (clades) based on type I enamel: *Equus hydruntinus*—*Equus gmelini* and *Equus latipes*—modern forms. It is logical that *Equus hydruntinus* and modern forms are located at opposite ends of the diagram as morphologically distinct taxa. The extinct *Equus gmelini* and *Equus latipes* are located in an intermediate position on this diagram. The former are morphologically similar to *Equus hydruntinus*, the latter to modern *Equus caballus* and *Equus przewalskii*. This structure of the scheme may, to some extent, reflect possible phylogenetic relationships within the studied groups. However, the placement of *Equus gmelini* Kamiana Mohyla remains problematic, and this may be due to geological age or taxonomic identification of specimens that may be close to modern forms.

**Author Contributions:** V.D. and L.R. conceptualized the research idea, designed the methodology, and collected and analyzed the data. M.B. conducted the statistical analysis and created diagrams. V.D. was responsible for funding acquisition, and both V.D. and L.R. were involved in the original draft preparation. All authors have read and agreed to the published version of the manuscript.

**Funding:** This work was supported by the International Visegrad Fund (52010462).

**Institutional Review Board Statement:** Not applicable.

**Data Availability Statement:** Data are contained within the article.

**Acknowledgments:** The authors appreciate the reviewers' and editor's valuable and profound comments.

**Conflicts of Interest:** The authors declare no conflict of interest.

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
