# Peer review of "Teeth Enamel Ultrastructural Analysis of Selected Equidae Taxa"

_diversity, doi:10.3390/d15111141_

Round 1
Reviewer 1 Report
Comments and Suggestions for Authors
All my suggestions are highlighted directly in the text.

Author Response
Dear Reviewer,
I want to extend my sincere appreciation for your invaluable comments and insights on my article. Your constructive feedback has been immensely helpful and has significantly elevated the scholarly quality of the work.
All your suggestions have been carefully considered, and the necessary revisions have been implemented. Please, find attached the PDF version of the article, which includes responses to each of your comments.
I hope these revisions meet your expectations, and I am grateful for the opportunity to enhance the overall quality of the manuscript based on your expert guidance.

Reviewer 2 Report
Comments and Suggestions for Authors
It has been a genuine pleasure to review the work you have undertaken. I look forward to following the progress of this enamel microstructure research line in future contributions.
Comments on the Quality of English LanguageIt would be beneficial to have a native English speaker review the language to enhance certain expressions. Otherwise, the work is easily understandable.
Author Response
Dear Reviewer,
Thank you very much for your kind words and positive feedback on our research. We greatly appreciate your support and interest in our work. We would like to inform you that we have expanded our research methods chapter to further enhance the quality and depth of the manuscript. We look forward to sharing our future contributions in this exciting field of enamel microstructure research. Thank you for your valuable input and your continued interest in our research.
Reviewer 3 Report
Comments and Suggestions for Authors
The study quantified enamel structures of molars of 5 different equid taxa from 7 localities, calculated corresponding K indices as morphometric descriptions and then examined the resulting datasets by means of statistical methods for their systematic and taxonomic signficance as well as paleoecological indicators. The sample covers both, recent and extinct taxa. As main result, the authors identify and confirm two clusters distinguishing between extinct and extant branches.
The statistical evaluation of the datasets is rigorous and yields interesting results. With the exception of a few typos, the study design as well as the presentation of the results are flawless and convincing.
Minor issues:
The Late Pleistocene is not capitalized throughout the manuscript. Please change 'late Pleistocene' to 'Late Pleistocene'.
Throughout the manuscript the phrase 'Equus przewalskii form Chornobyl exclusion zone' is used. Actually, I am unsure, whether the 'form' should not read 'from'. Please check.
Page 4, lines 161, 169, 174, 181: There is a weird symbol in the test statistics which is probably due to a pdf mismatch. Please change.
Page 4, lines 185-187: Interpunctation is at odds in the first phrase of this paragraph. Please correct.
Page 5, line 207: 'top and bottom molar teeth' should read 'upper and lower molars'. Please correct.
Comments on the Quality of English Language
Except of minor issues, there is no problem with the language.
Author Response
Dear Reviewer,
Thank you for your thorough review and interest in our work. We would like to inform you that all the points reviewers raised have been addressed and corrected in our article. Below are responses to your specific comments:
- "late Pleistocene" has been changed to "Late Pleistocene" as suggested.
- The phrase "Equus przewalskii form Chornobyl exclusion zone" has been corrected to "Equus przewalskii from Chornobyl exclusion zone."
- The unusual symbols in the statistics on page 4 have been eliminated.
- Punctuation on page 4, lines 185-187, has been revised.
- "top and bottom molar teeth" has been changed to "upper and lower molars" as per your suggestion.
Once again, we appreciate your review and your feedback has been valuable in improving the quality of our manuscript.
Round 2
Reviewer 1 Report
Comments and Suggestions for Authors
None